# Gestational Bell’s Palsy Is Associated with Higher Blood Pressure during Late Pregnancy and Lower Birth Weight: A Retrospective Case-Control Study

**DOI:** 10.3390/ijerph181910342

**Published:** 2021-09-30

**Authors:** Sumonthip Leelawai, Chitkasaem Suwanrath, Nannapat Pruphetkaew, Pensri Chongphattararot, Pornchai Sathirapanya

**Affiliations:** 1Division of Neurology, Department of Internal Medicine, Faculty of Medicine, Prince of Songkla University, Hat Yai, Songkhla 90110, Thailand; nan.sumonthip@gmail.com (S.L.); cpensri@medicine.psu.ac.th (P.C.); 2Department of Obstetrics and Gynecology, Prince of Songkla University, Hat Yai, Songkhla 90110, Thailand; schitkas@medicine.psu.ac.th; 3Epidemiology Unit, Faculty of Medicine, Prince of Songkla University, Hat Yai, Songkhla 90110, Thailand; pnannapa@medicine.psu.ac.th

**Keywords:** Bell’s palsy, eclampsia, preeclampsia, pregnancy, hypertension, diabetes

## Abstract

The associations between gestational Bell’s palsy (GBP) and late obstetric complications (LOCs), i.e., preeclampsia (PE), eclampsia (EC), gestational hypertension (GHT), and gestational diabetes mellitus (GDM) remain unclear. This study aimed to evaluate these associations and the neonatal health of the newborns born from pregnant women with and without GBP. A retrospective 1:5 case-control study matching exact maternal age and gravidity between pregnant women with and without GBP in Songklanagarind Hospital from 2006 to 2016 was conducted. The associations between GBP and PE, EC, GHT, and GDM, as well as comparison of the newborns’ health indices were analyzed by bivariate analysis (*p* < 0.05). Eight GBP cases out of 8,756 pregnant women were recruited. Six GBP cases were first or second gravid. GBP occurred during the third trimester in five cases. Except for higher median systolic blood pressure (125 (114.2, 127.5) vs. (110 (107.0, 116.0), *p* = 0.045) and diastolic blood pressures (77 (73.0, 80.8) vs. 70 (65.0, 73.2), *p* = 0.021) in the GBP cases, associations between GBP and all LOCs could not be concluded due to the lack of power. However, a significantly lower mean birth weight in the newborns of GBP mothers was found (2672.2 (744.0) vs. 3154.8 (464.7), *p* = 0.016) with statistically significant power. Except for the higher blood pressures and lower birth weights of the newborns of GBP mothers, an association between GBP and LOCs remains inconclusive.

## 1. Introduction

The pathogenesis of Bell’s palsy has not been clearly explained. Reactivation of the latent herpes simplex virus (HSV) infection in the geniculate ganglion of the facial nerve and ischemic facial neuropathy associated with hypertension or diabetes mellitus has been postulated [1]. Bell’s palsy occurring during pregnancy, called “gestational Bell’s palsy (GBP)” in this study, commonly takes place during the late third trimester or in the immediate postpartum period, which is also the same onset time of preeclampsia (PE) or eclampsia (EC). Based on the same onset time of these conditions, it has been hypothesized that they might share a common pathogenesis [2]. Moreover, GBP is considered a risk of PE or EC too [3,4,5,6]. This presumption is based on only a few case reports, case series, and unmatched-case studies. The hypothesis of this study is that GBP is not related to these pregnancy complications. Therefore, this study aimed to investigate whether there are significant associations between GBP and late obstetric complications (LOCs), e.g., PE, EC, and other pregnancy-related complications, i.e., gestational hypertension (GHT) and gestational diabetes mellitus (GDM), in a matched-case study design. The difference in neonatal health indices between the newborns born from GBP (GBP+) and non-GBP (GBP−) mothers were evaluated as well.

## 2. Materials and Methods

### 2.1. Study Participants, Design, and Setting

This was a retrospective 1:5 case-control study matching the exact maternal age and the order of gravidity at the presentation of GBP between GBP+ and GBP− pregnant women. The newborns of GBP+ and GBP− pregnant women were matched corresponding to their mothers, and neonatal characteristics were analyzed for significant differences. We retrospectively enrolled the study cases from pregnant women who completed their pregnancies and delivered alive babies between January 2006 and December 2016 in Songklanagarind Hospital, a 900-bed referral and medical teaching hospital affiliated with Prince of Songkla University in southern Thailand. We defined GBP+ cases as pregnant women who acquired Bell’s palsy during their pregnancies and until six weeks postpartum. We used the Random.org website to randomly select controls from GBP− pregnant women in the hospital patient information database after matching the maternal age and order of gravidity of controls to those of individual case. All the enrolled pregnant women had regular follow-ups and complete obstetric history records in our center. We excluded pregnant women with a known history of essential hypertension or overt diabetes mellitus, or both, before the index pregnancies.

### 2.2. Terms and Definitions

The definitions of medical terms used in this study were as follows:

Bell’s palsy was defined by an acute unilateral peripheral facial paralysis without an identifiable cause [1];

Gestational hypertension (GHT) was an elevation of systolic blood pressure (SBP) ≥140 mmHg or diastolic blood pressure (DBP) ≥90 mmHg measured after 20 weeks of gestation without proteinuria in a previously normotensive woman, with the higher SBP and DBP normalizing by 12 weeks postpartum [7];

For gestational diabetes mellitus (GDM), we followed the diagnostic criteria of the American College of Obstetricians and Gynecologists as follows: if the one-hour plasma glucose level after a 50-g oral glucose challenge test (1 h 50-g GCT) was ≥140 mg/dL, then a three-hour 100-g oral glucose tolerance test (3 h 100-g OGTT) would be conducted. A diagnosis of GDM was made when two or more of the plasma glucose levels met the diagnostic levels as follows: (a) fasting plasma glucose ≥95 mg/dL; (b) 1st-hour plasma glucose ≥180 mg/dL; (c) 2nd-hour plasma glucose ≥155 mg/dL; or (d) 3rd-hour plasma glucose ≥140 mg/dL [8];

Preeclampsia (PE) was a pregnancy with a SBP ≥140 mmHg or DBP ≥90 mmHg and at least one of the following features: a) 1+ proteinuria on a urine dipstick; b) 24-h urine protein >300 mg; or c) a urine protein-to-creatinine ratio ≥0.3 [7];

Eclampsia (EC) was defined as the clinical features of PE plus seizures that were not attributable to other causes [7];

Low birth weight was diagnosed when birth weight was <2500 g [9].

### 2.3. Data Collection

GBP+ and GBP− patients’ demographic data, obstetric profiles, and gestational age at the onset of GBP were retrieved from the hospital computerized medical records. The presence of PE, EC, GHT, and GDM was carefully identified and confirmed by one of the authors (Suwanrath C.) based on the diagnostic criteria mentioned. We specifically recorded the maternal SBP and DBP of GBP+ cases at the time of GBP occurrence, and those of GBP− cases by averaging the two values of each parameter recorded at a comparable gestational age (±2 weeks). We also collected characteristics of the newborns of the GBP+ and GBP− mothers, including gestational age at birth, birth weight, body length, Apgar scores, and route of delivery, for comparative analysis.

### 2.4. Statistical Analysis

Descriptive statistics were used to compare the baseline characteristics between the GBP+ and GBP− pregnant women. A Wilcoxon rank-sum test, *t*-test, and Fisher’s exact test were used for the evaluation of significant differences (*p* < 0.05) in the characteristics between the two mother groups and their corresponding newborns. Associations between GBP and PE, EC, GHT, and GDM were analyzed by bivariate analysis (*p* < 0.05).

### 2.5. Ethical Considerations

The study protocol was approved by the Ethics Committee of the Faculty of Medicine, Prince of Songkla University (EC Protocol Registration No. 60-132-14-4). We strictly followed the regulations in the 1964 Declaration of Helsinki and its following amendments. We also followed the current good clinical practice guidelines in doing this research. We completely anonymized the identifiable personal information of the study cases.

## 3. Results

There were eight out of 8,756 (0.09%) pregnant women who met the inclusion criteria of GBP+ during the study period. Forty GBP− pregnant women were randomly selected as controls using the mentioned website from the hospital database. The median (IQR) maternal age at the onset of GBP was 31.5 (26.8, 36.2) years. GBP occurred during the first (three cases), second (three cases), fourth (one case), and fifth (one case) gestations. The onset of GBP was in the late second trimester in two cases, the third trimester in five cases, and postpartum in one case. Among the GBP+ pregnant women, we identified only one case with PE, and one other case with both GDM and GHT. No case associated with EC was found.

We found only a significantly higher median (IQR) SBP and DBP in the GBP+ group by bivariable analysis. However, both the higher blood pressures did not meet the diagnostic criteria of GHT. The median (IQR) of 50-g GCT in the GBP+ group was non-significantly higher than that in the GBP− group. The SBP, DBP, and 50-g GCT had statistically significant power in this study. Associations between GBP and GHT, GDM, PE, and EC were unable to be concluded due to the lack of power (Table 1).

In comparison of the neonatal health characteristics, except for a significantly lower mean birth weight in the newborns of GBP+ mothers, there were no significant differences in gestational age at birth, body length, Apgar scores at 1 and 5 min, or route of delivery. A statistically significant power of low birth weight was also found (Table 2). The indications of cesarean sections (CS) in our GBP+ cases were: previous CS in labor in three cases; elective CS in two cases; severe preeclampsia in one case; and fetal distress in one case.

Because of the very small number of GBP+ cases, we could not perform a multivariate analysis.

## 4. Discussion

In contrast to the previous studies, only a significantly higher SBP and DBP in GBP+ pregnant women were found, but the level of blood pressures did not fulfil the diagnostic criteria for GHT. As most of the previous studies reporting a higher risk of PE among pregnant women with GBP are based on merely a few case reports or small case series, the association between GBP and PE remains doubtful. To date, there have been few studies enrolling large number of GBP cases to confirm the associations. From a search of the literature, we found two studies which proposed that GBP was “probably” associated with GHT, PE, and other pregnancy-related adverse events. For example, one study found that GBP+ pregnant women experienced a six-fold higher incidence of PE and a 1.5-fold higher incidence of GHT compared to the expected rates of both conditions in the general female population [3]. Another study reported significantly higher rates of severe PE and cesarean deliveries, but favorable neonatal outcomes in the GBP+ group [4]. It is noteworthy that although both studies included larger number of GBP+ cases than the earlier studies, the former study compared the actual incidences of PE and GHT obtained from the study with the “expected” incidences among the “general” (non-pregnant) female population from a nationwide survey reported in another study, while the latter was an unmatched-case study between GBP+ and GBP− pregnant cases. Therefore, association between GBP and PE or GHT has not been confirmed based on the available data.

It has been hypothesized that the incidence of Bell’s palsy is higher during pregnancy. However, Vabrec, et al. [10] argued against the higher incidences of Bell’s palsy during pregnancy reported in the previous studies. They suggested that the exaggeration in enrollment of controls, plus active recruitment of pregnant women with GBP which were scarcely found in real practice, were possible causes of the aggregation of GBP cases and overestimation of the incidence of GBP among pregnant women. Another argument was the comparison of the incidences in which different populations were used as the nominators for calculating the incidences between cases and controls. For example, the reported incidence of GBP in 45:100,000 “live births” compared with 17:100,000 “women of child bearing age” by Hilsinger, et al. [11] was arguable. A recent national cohort case-control study from Korea also reported that pregnancy did not increase the incidence of Bell’s palsy [12]. Based on the arguments of Vabrec et al. on this issue and the recent Korean study mentioned, the previous understanding that the incidence of Bell’s palsy during pregnancy is higher than in general female population seems inconclusive.

Although the higher incidence of GBP than Bell’s palsy in general females is controversial, some case reports and small case series associated the occurrence of GBP with PE, EC, GHT, and GDM. How these disorders are related has been a question due for an elucidation. A consideration of the proposed pathogeneses of these LOCs in relation to those of Bell’s palsy would be helpful. Generally, the definite pathogenesis of Bell’s palsy has not been clearly elucidated. The main hypothesis currently is that Bell’s palsy is caused by the reactivation of a latent HSV infection in the geniculate ganglion of the facial nerve. The other proposed pathogeneses are: (1) ischemic facial neuropathy complicated with long-standing systemic hypertension, glucose intolerance, or diabetes mellitus; (2) altered host immunity predisposing the patients to the reactivation of a latent HSV or other viral infections; and (3) immunologically inflammatory demyelination of the facial nerve, such as a variant form of Guillain–Barre syndrome [1]. Whether GBP and LOCs share a proposed pathogenesis of immune-mediated vasculopathy of facial nerves and placentas has not been clarified [13,14].

The significantly lower mean birth weight in the newborns of GBP+ mothers was the only abnormal neonatal health index found with statistically significant power in our study. As the gestational age at birth of the newborns was not different between the two groups (Table 2), the lower mean birth weight of newborns in the GBP+ group was not due to premature birth. We postulate that it possibly results from placental vascular insufficiency in the fetuses born from GBP+ mothers. Since monitoring of intrauterine fetal growth was not done in this study, intrauterine growth retardation cannot specifically be addressed.

The specific parameter-matched, case-control study design was a methodological strength of our study. We matched the exact maternal age and the order of gravidity at the occurrence of GBP between cases and controls to eliminate the variation of immune responses during pregnancies, which predispose to the development of LOCs. The paucity of GBP cases in real practice precludes a study with large number of GBP cases enrollable, which limits our conclusion strength. However, the significantly higher SBP, DBP, and lower birth weight, with the statistically significant power found in our study, imply that GBP is associated with higher blood pressure in GBP+ pregnant women and lower birth weights in the newborns. Future studies with larger number of GBP cases are required to re-evaluate our current findings. Eventually, we would like to suggest that as age-dependent physiological factors and immunological changes in subsequent orders of pregnancy occur, both maternal age and gravidity should be considered as matching parameters in the future studies.

## 5. Conclusions

Higher SBP and DBP during late pregnancy are associated with GBP. Lower birth weight of the newborns of GBP+ mothers are also associated with GBP in our study, which has not been reported earlier. Further studies with a larger sample size, to confirm the association of GBP with LOCs, and to elucidate the pathogenesis of GBP in association with LOCs, are required.

## Figures and Tables

**Table 1 ijerph-18-10342-t001:** Comparison of maternal characteristics and late pregnancy complications between pregnant women with and without gestational Bell’s palsy (GBP+ and GBP−) by bivariate analysis.

Variable	GBP+ Cases (*n* = 8)	GBP− Cases (*n* = 40)	*p*-Value	Power
*General characteristics*				
Age, years, median (IQR)	31.5 (26.8, 36.2)	31.5 (26.8, 36.2)	1	
GPA, n (%)			1	
G1	3 (37.5)	15 (37.5)		
G2P1	3 (37.5)	15 (37.5)		
G4P3	1 (12.5)	5 (12.5)		
G5P4	1 (12.5)	5 (12.5)		
*Association with GDM, GHT,* *Preeclampsia and Eclampsia* 1 h 50-g GCT, mg/dL, median (IQR)	144.0 (95.0, 147.0)	130.0 (116.0, 146.5)	1	1.00
SBP, mmHg, median (IQR)	125 (114.2, 127.5)	110 (107.0, 116.0)	0.045 *	1.00
DBP, mmHg, median (IQR)	77 (73.0, 80.8)	70 (65.0, 73.2)	0.021 *	1.00
GDM, n (%)			0.429	0.11
Yes	1 (12.5)	2 (5.0)		
No	7 (87.5)	38 (95.0)		
GHT, n (%)			0.167	0.46
Yes	1 (12.5)	0 (0)		
No	7 (87.5)	40 (100)		
Preeclampsia, n (%)			0.167	0.46
Yes	1 (12.5)	0 (0)		
No	7 (87.5)	40 (100)		

Abbreviations: GPA; gravida/para/abortus (GPA) system, 1 h 50-g GCT; one-hour, 50-g glucose challenge test, SBP; systolic blood pressure, DBP; diastolic blood pressure, mmHg; millimeters of mercury, GDM; gestational diabetes mellitus, GHT; gestational hypertension, SD; standard deviation, IQR; interquartile range. * *p* < 0.05, Wilcoxon rank-sum test.

**Table 2 ijerph-18-10342-t002:** Comparison of neonatal health characteristics between the newborns of pregnant women with and without gestational Bell’s palsy (GBP+ and GBP−) by bivariate analysis.

Variable	GBP+ (*n* = 9 ^a^)	GBP− (*n* = 40)	*p*-Value	Power
Gestational age at birth, wks., median (IQR)	38.0 (37.0, 39.0)	38.5 (38.0, 40.0)	0.077	0.24
Birth weight, g., mean (SD)	2672.2 (744.0)	3154.8 (464.7)	0.016 *	1.00
Birth length, cm., median (IQR)	49.0 (48.0, 54.0)	50.0 (49.0, 51.0)	0.927	0.71
Apgar 1, median (IQR)	9 (8, 9)	9 (9, 9)	0.438	0.05
Apgar 5, median (IQR)	10 (9, 10)	10 (9, 10)	0.917	0.05
Route of delivery, n (%)			0.064	0.53
Cesarean section	7 (77.8)	16 (40)		
Normal labor	2 (22.2)	24 (60)		

^a^ = one pair of twins. Abbreviations: wks; weeks, g; gram(s), cm; centimeter(s), SD; standard deviation, IQR; interquartile range, Apgar 1; Apgar score at 1 min after birth, Apgar 5; Apgar score at 5 min after birth. * *p* < 0.05, chi-squared test.

## Data Availability

All the data and analysis methods were reported in this manuscript.

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
