# Peer review of "Gestational Bell’s Palsy Is Associated with Higher Blood Pressure during Late Pregnancy and Lower Birth Weight: A Retrospective Case-Control Study"

_ijerph, 2021, doi:10.3390/ijerph181910342_

Round 1
Reviewer 1 Report
Comments to the paper:
Gestational Bell’s palsy is not associated with preeclampsia, eclampsia or other late obstetrics complications, but low birth weight: a retrospective case-control study.
This paper describes simple associations between Gestational Bell’s Palsy (GBP) and some perinatal outcomes in a case-control study. Samples size only allowed bivariate analysis, making findings informative, but difficult to interpret.
Specific comments.
- Information on gestational age at birth is needed, to better interpret birth weight association with GBP.
- The use of BMI in pregnant women is not recommended because placenta, fetus and amniotic fluid distort interpretation of this anthropometric variable. Instead of that, please provide total or partial gestational weight gain and adequacy interpretation. This may add some information on the nutritional status of women and adiposity which may be also discussed as contributors to the mechanisms of GBP. If available, information on other markers of the nutritional and/or metabolic status of women must be provided (hemoglobin, lipid levels, etc.).
- A correction is needed in Table 1 and 2 and lines 104-105. Statistical evaluation of association was done by bivariate analysis.
Reviewer 2 Report
Line 85 - need to clarify that this is their definition for the purpose of the study, or should update definition in line with currently accepted definitions and cite reference for this (eg ACOG, NICE)
Line 139 and Lines 236-237- Low birth weight? Evidence for this statement? Did you consider gestation differences in groups? What was definition of low birth weight? Would the use of centiles be appropriate? Any evidence of IUGR from prenatal scans/abnormal fetal dopplers for example? Was there iatrogenic preterm birth?
Table 2 - route of delivery - seems to be a very high rate of caesarean birth in the bells palsy group - what were the indications for CS? Rate of emergency versus planned CS?
line 173-184 - what is the rate of bells palsy in your local/general population?
line 212 - evidence for this statement? Why is this the most reasonable explanation? Was there evidence of HSV or other immunocompromise in your case series?
References - seems to be a very limited number, and does not include most recent articles eg.
Pregnancy does not increase the risk of Bells palsy: a national Cohort study 2020 Jan;41(1):e111-e117.
doi: 10.1097/MAO.0000000000002421. From the Korean group – larger numbers of BP cases
You haven't referenced or eluded to your previous publication of some of these cases -
Bells Palsy in pregnancy: A case series. 2020 Nov 26;12(3):452-459.
doi: 10.1159/000509682. eCollection Sep-Dec 2020.
Reviewer 3 Report
This study contains novel data regarding the association between gestational Bell's palsy and late obstetric complications as well as newborn health indices.
Introduction
-Lines 43-44 Due to the same onset time of these conditions, it has been believed that they might share a common pathogenesis- Are there any other similarities between these conditions apart from the time period?
-Line 44 Moreover, GBP is possibly a risk of PE or EC- What sort of risk are you referring to?
-Lines 45-46 This presumption is based on only a few case reports, case series and unmatched-case studies. - Supporting references are needed here.
-Can you explain why this study is needed? What evidence gap is it filling?
Methods
-Was randomisation concealed?
-Was the study limited to singleton pregnancies?
-Who performed and confirmed the diagnosis of the clinical conditions amongst the participants?
-Lines 91-92: GBP+ and GBP- patients’ demographic data, parameters of metabolic disorders, obstetric profiles, and gestational age at the onset of GBP were recorded.- How?
-weight, height, body mass index (BMI), SBP and DBP of GBP+ cases at the time of GBP occurrence- and GBP+ and GBP- including birth weight, body length, Apgar scores and route of delivery for comparative analysis. How were these variables measured? What equipment/personnel was used?
Results
-Table 1- weight, height and BMI- Is that pre-pregnancy? What timepoint is blood pressure?
-Table 2- For caesarean section and normal labor- I am assuming that the n(%) is reported?
Discussion
-Only significant higher SBP and DBP in GBP+ pregnant women were 154
found, but the level of elevated blood pressures did not fulfil the diagnostic criteria for GHT- Have any other studies reported elevated BP only? Was blood pressure measured at one time point?
-Line 173: It has been believed that incidence of BP is higher during pregnancy. - A supporting reference is needed here.
-Lines 235-236: The significant lower mean birth weight in the newborns of GBP+ mother was the only abnormal neonatal health index found in our study. We postulate that it is possibly resulted from placental vascular insufficiency in the fetuses born from GBP+ women. - Greater discussion of this finding is needed compared to the length discussion of the pathogenesis of BP which has not been analysed in this study.
Reviewer 4 Report
- Summary: This is a case control study with cases selected from deliveries between January 2006 and 2016. Eight cases among 8756 deliveries were noted. Controls were randomly selected from the electronic medical record database controlling for maternal age and parity with 5 controls selected for each case. The authors found that most cases occurred in the second and third trimester of pregnancy, cases had lower mean birth weight and although not significant, had higher CD rates. There was no difference in the proportion of women with gestational hypertension, gestational diabetes or preeclampsia. The authors conclude that there is no relation between the development of Bell’s palsy in pregnancy and the occurrence of these late obstetrical complications.
- Strengths: are the number of cases and strict methodology followed.
- The perceived limitations are: the study is probably not powered to analyze the differences in stated outcomes between cases and controls. Study design may require analyzing women with and without the outcome of preeclampsia or any of the other stated outcomes and looking at the frequency of the rare outcome of Bell’s Palsy between the two groups.
- Limited clinical importance. The authors report lower birthweight and higher cesarean section rates, may want to comment on those outcomes.
- Comments on individual sections
Title: appropriate
Abstract adequate, no discrepancies.
Introduction is adequate and succinct.
Materials and Methods
- Page 2 line 61: If controls were selected 5:1 matched based on parity and maternal age, when was randomization performed? If the implication is that women were randomly selected after setting limits for age and parity, then this sentence would need to be rephrased.
- Page 2 line 69-88: References needed for these paragraphs.
- Page 2 line 75: It seems that at least two different methods were used to diagnose GDM, this is not central to this manuscript but should be stated clearly.
- This study might benefit from a power analysis to determine if it was powered to find a difference between the cases and controls for all outcomes analyzed.
Is the study design appropriate to allow their hypothesis to be tested? - There is no stated hypothesis
Results:
- Page 3 line 115: I think this means that 40 women were randomly selected...
- Table 1: No cases of eclampsia to include in this table, consider removing from table and not be an outcome variable. A comment can be inserted in text.
- Table 1: Cesarean section: Include some comment on the indications for CD among women with Bell’s Palsy
For original research papers, is the hypothesis clearly stated? The aims are clearly stated.
Discussion:
- Page 5 line 156-157: Although there are 8 cases in this series, overall the number is small to establish the risk for a condition that appears in 5-7% of the population. Even if the analysis were to be performed in the opposite direction, the number of cases of Bell Palsy among women with or without preeclampsia or GHTN would require a large number of cases given the incidence of Bell's Palsy (0.09% per this report). Due to the low frequency of the condition, a case control model to determine predictors or associations with a rare condition is useful when the predictor or condition has a relatively high incidence in both groups. In this case, birth weight was available for all cases and controls and found to be different. This is also true for mode of delivery, which was available for all cases.
- Page 5 line 170: This case control study is not powered to make this statement. The association is neither confirmed nor discarded.
- Page 5 line 178-179: Please rephrase this statement, it is difficult to understand.
- Page 6 line 196-204: Not sure how this is relevant to this discussion.
Conclusion:
- Page 6 line 252-254: This has not been demonstrated in this study.
Round 2
Reviewer 1 Report
None
Author Response
Reviewer 1 comment -Round 2
Response
Comments and Suggestions for Authors
None
Response: Thank you for your kind support and improvement of my work.
Reviewer 2 Report
Thank you for your attention to the previous review recommendations. I have a couple of minor points to add:
Lines 146-148 "elective GA at proper gestational age" , I note this has been added in response to my previous feedback. If the indication for elective caesarean section is previous caesarean section, and the average gestation was 38 weeks, this required further comment and explanation. ACOG and RCOG guidelines recommend that elective CS should be done at 39+ weeks to reduce neonatal respiratory complications. Perhaps this study highlights a local practice that differs from other centres around the world. GBP in itself would rarely be an indication for delivery, and I can see no reason why it would be an indication for delivery by CS in itself.
Line 253 - language needs review, can I suggest replace "did not implement" with "did not occur"
Author Response
Reviewer 2 comment-Round 2
Comments and Suggestions for Authors
Thank you for your attention to the previous review recommendations. I have a couple of minor points to add:
Lines 146-148 "elective CS at proper gestational age", I note this has been added in response to my previous feedback. If the indication for elective caesarean section is previous caesarean section, and the average gestation was 38 weeks, this required further comment and explanation. ACOG and RCOG guidelines recommend that elective CS should be done at 39+ weeks to reduce neonatal respiratory complications. Perhaps this study highlights a local practice that differs from other centres around the world. GBP in itself would rarely be an indication for delivery, and I can see no reason why it would be an indication for delivery by CS in itself.
Response: Thank you for kind support and comments. Cesarean section is commonly done at gestational age of 39 wks in our practice as an elective case, unless an indication presents. After reviewing the indications of CS in our cases, we added them to the manuscript text. (Line 143-145)
Line 253 - language needs review, can I suggest replace "did not implement" with "did not occur"
Response: Thank you. May I rephrase the sentence using “was not done” to replace "did not implement". as follow: Since monitoring of intrauterine fetal growth was not done in this study, intrauterine growth retardation cannot specifically be addressed.
Reviewer 4 Report
Summary: This is a case control study with cases selected from deliveries between January 2006 and 2016. Eight cases among 8756 deliveries were noted. Controls were randomly selected from the electronic medical record database controlling for maternal age and parity with 5 controls selected for each case. The authors found that most cases occurred in the second and third trimester of pregnancy, cases had lower mean birth weight and although not significant, had higher CD rates. There was no difference in the proportion of women with gestational hypertension, gestational diabetes or preeclampsia. The authors conclude that there is no relation between the development of Bell’s palsy in pregnancy and the occurrence of these late obstetrical complications but these results are not statistically significant as it is not powered to detect these differences.
This is a review of the second version of this manuscript:
The following observations are made:
- The title is misleading, the patient’s with Bell’s palsy have a higher blood pressure but this cannot be stated as elevated blood pressure. Elevated blood pressure would imply hypertension and these were not hypertensive women.
- Page 2 line 63-64: Based on the description of the work performed, controls were not RANDOMIZED, but rather randomly selected while matching for maternal age and parity.
- Page 2 line 84-88: The authors define criteria for diagnosis of overt diabetes based on criteria used on non-pregnant individuals. However, these women were presumably excluded from this study based on the exclusion criteria listed under section 2.1
- Please do not use the acronym BP for Bells Palsy as this is most often used to signify “Blood Pressure”.
- When mentioning the term “power analysis” please use the proper terminology. There is no such thing as low power analysis, but rather ”lack of power” or “not statistically significant”
- In the discussion please focus on your results. The discussion is rather long and addresses issues not investigated in this study. The main results were lower birthweight and higher blood pressure but not elevated blood pressure, GHTN or preeclampsia.
- Page 6 lines 209-223: this section is not relevant to the study aims, study results or study findings. You might want to consider removing.
- Page 6 lines 225-245: None of these hypotheses are based on the data presented and the data presented does not support any of these hypotheses.
- Page 6 lines 233-240: The data you report does not support this statement.
- On page 7, line 258-260 the discussion talks about how matching would avoid the effect that hypertension or diabetes could have on outcomes. This is confusing as women with hypertension or diabetes would have ben excluded as state in the methods section.
- The conclusion should be modified as the blood pressure was not elevated, it was higher but did not meet criteria for hypertension. The statistical analysis comparing the blood pressure between the groups was the Wilcoxon Rank Sum test, which is inappropriate. The proper test would have been student T-test.
Author Response
Please see the attachment

This manuscript is a resubmission of an earlier submission. The following is a list of the peer review reports and author responses from that submission.